# Population Genetic Divergence and Environment Influence the Gut Microbiome in Oregon Threespine Stickleback

**DOI:** 10.3390/genes10070484

**Published:** 2019-06-26

**Authors:** Robert A. Steury, Mark C. Currey, William A. Cresko, Brendan J. M. Bohannan

**Affiliations:** Department of Biology, Institute of Ecology and Evolution, University of Oregon, Eugene, OR 97403-5289, USA

**Keywords:** host-bacterial associations, microbiome, microbiota, population genomics, genetic divergence, ecology

## Abstract

Much of animal-associated microbiome research has been conducted in species for which little is known of their natural ecology and evolution. Microbiome studies that combine population genetic, environment, and geographic data for wild organisms can be very informative, especially in situations where host genetic variation and the environment both influence microbiome variation. The few studies that have related population genetic and microbiome variation in wild populations have been constrained by observation-based kinship data or incomplete genomic information. Here we integrate population genomic and microbiome analyses in wild threespine stickleback fish distributed throughout western Oregon, USA. We found that gut microbiome diversity and composition partitioned more among than within wild host populations and was better explained by host population genetic divergence than by environment and geography. We also identified gut microbial taxa that were most differentially abundant across environments and across genetically divergent populations. Our findings highlight the benefits of studies that investigate host-associated microbiomes in wild organisms.

## 1. Introduction

Microbiome research is now a major area of research in ecology and evolution [1], which is partly due to an increased understanding of the importance of the microbiome to animal fitness [2]. Recent microbiome research has frequently focused on the vertebrate gut, for several reasons. The densely colonized and metabolically diverse gut microbiome can influence host nutrition [3] and development [4], including the onset of host immunity [5]. Furthermore, a normal gut microbiome is important in preventing pathogen infection and many host disorders, such as inflammatory bowel disease and colorectal cancer in humans [6,7,8]. 

Despite this previous research, we still do not fully understand how host genetics and environmental factors interact to influence the structure and function of an individual animal’s microbiome [9,10]. During the earliest stages of development, most vertebrate hosts are essentially sterile, but then enter a world dominated by microbes [11], where they are rapidly colonized [12]. This initial colonization can influence the host organism and its associated microbiome throughout its life [13]. Microbial community structure in non-host environments can be predicted by variation in environmental factors [14,15]. For instance, salinity gradients and fluctuations in temperature and light are influential factors in aquatic environments [16,17]. Therefore, a common hypothesis is that the primary factors influencing microbiome variation across hosts are environmental in nature. Indeed, studies have documented that host-associated microbiomes can be influenced by environmental factors [18,19]. However, in some cases microbial taxa found associated with the host are not commonly found in its external environment, suggesting that host selection and other processes (e.g., host-to-host transmission and stochasticity) must also play roles.

Given the extensive effects the microbiota can have on the host, it is not surprising that the host has evolved to modulate the microbiota. The innate and adaptive immune systems evolve to monitor and alter the abundances of microbes [1]. Like other organismal traits, these can be genetically determined, and therefore can be underpinned by genetic variation. For instance, this is inferred from observations of divergence of major gut bacterial phyla between human and animals [20] and microbiome similarities among genetically related humans [10,21,22,23]. Moreover, microbial quantitative-trait loci mapping (mbQTL) and genome-wide microbial association studies conducted in mice [23,24] and humans [25,26] revealed that important underlying host genetics are complex and possibly heritable. We do not yet understand the full complexity of these traits, nor how environmental variation and segregating host genetics (as compared to induced mutations) interact to modulate the microbiome in wild hosts.

The majority of the research on host-microbiome associations has focused on model organisms in the laboratory. Studying microbiome associations in wild hosts could augment and expand what is learned in lab settings by incorporating natural genetic variation interacting with ecologically relevant environmental variation, often in environments in which the hosts evolved. A historical problem has been a lack of the genetic and genomic tools for such studies on non-model organisms, but this problem is quickly receding because of advances in genomic sequencing technology (e.g., Restriction site-associated DNA sequencing (RAD-seq)) [27]. These tools allow for the fine-scale determination of the relationship among individuals and populations in the wild through measurements of genome-wide genetic diversity using next-generation sequencing approaches [28]. These could then be used, for example, to precisely quantify the genetic distance among populations and relate it to microbiome, geographic, or environment variation. 

An excellent model for studying the combined influences of environment and genetics on microbiome variation in the wild is the threespine stickleback fish (*Gasterosteus aculeatus*) [29]. This small teleost fish has a thoroughly studied evolutionary and ecological history in numerous locations throughout the Holarctic [29]. Stickleback are abundant and nearly ubiquitous in coastal regions of the Northern Hemisphere where they display marine, anadromous, and freshwater forms that themselves are phenotypically and genetically diversified. A fully sequenced and annotated genome and population genomic tools permit broad genomic sequence resolution and powerful inference of relatedness among populations and individuals [30]. Moreover, this species is amenable to laboratory manipulations and is tractable for microbiome research. For instance, differences in gene transcription have been compared between “gnotobiotic” stickleback (with a known microbiome) raised in either conventional or germ-free lab environments [31]. Gnotobiotic oceanic and freshwater stickleback have also been tested for varying immune response to microbiome exposure [32]. In addition, wild stickleback populations have been used to study how environment and host diet, sex, immune genes, and population level genetic relatedness and diversity relate to the gut microbiome [33,34,35,36]. Previous research in wild stickleback populations found that genetically divergent host populations that inhabit the same watershed, or in estuaries near the watershed, exhibited more divergent gut microbiota [36]. 

Here we expand upon this work by incorporating host whole-genome data and populations from multiple geographically and environmentally diverse watersheds. We build upon years of foundational work on stickleback in the wild by deeply sequencing both the host genome and the gut bacterial communities associated with genetically divergent wild populations of stickleback drawn from two differing natural environments (estuary and freshwater). We found that microbiome composition was better predicted by fish population genetic divergence than by geographic distance and environment. Furthermore, we found evidence that these differences among populations could be the result of differences in the relative abundance of a subset of the gut microbiome in these fish.

## 2. Materials and Methods 

### 2.1. Sample Collections

We collected six wild threespine stickleback populations for the diversity of the environments that they inhabit, known population structure, and geographic distribution in Oregon. One hundred and twenty-six fish of assumed mixed sex were collected between June and August in 2015 and 2017 in western Oregon, USA. Collections included ≥ 20 fish from each of six sites varying in distance apart from one another among three major watersheds (Figure 1 and Table 1). We collected pairs of estuarine and freshwater populations found in two separate coastal watersheds. One pair, from the Siuslaw watershed, were collected from Cushman Slough and Lily Lake, and the other, from the Umpqua watershed, were collected from Dean Creek and Eel Creek (Table 1). We also collected stickleback at two inland freshwater sites, Green Island and Lynx Hollow, located ~ 300 river miles inland in the Willamette Basin. These populations are from the main stem and Coast Fork of the Willamette River watershed, respectively. All collections were carried out as in our previous work according to approved University of Oregon, institutional animal care and use committee (IACUC) protocols, and under Oregon Department of Fish and Wildlife scientific taking permit numbers 19122 and 20770 [30,37].

### 2.2. Gut Isolation and Soma Preservation and DNA Extraction for Genomic Analysis

Fish were euthanized with an IACUC-approved lethal dose of MS222. Guts were dissected in the field and flash frozen immediately in liquid nitrogen, and then stored −80 °C. Somas for each fish were preserved in 95% ethanol. Salinity (via refractometer) and temperature at each site were measured at the time of sampling. DNA was extracted from fin clips (~20 mg) from ethanol-stored whole fish using a Blood and Tissue Kit (Qiagen, Venlo, Netherlands). Quantification of DNA concentration was performed on a Qubit fluorometer (Invitrogen, Waltham, MA, USA) using the double strand DNA (dsDNA) broad range assay kit. DNA dilution plates for each population were created by diluting to a concentration of 25 ng/ul using EB in 96 deep-well plates.

### 2.3. Restriction Site-Associated DNA Sequencing (RAD) Library Construction and Single-Nucleotide Polymorphism Discovery and Genomic Analysis

RAD-seq libraries were created as in previous work [27,30]. Briefly, for each population sequenced, the endonuclease *SbfI-HF* (New England Biolabs, Waltham, MA, USA) was used to cut DNA at specific locations spread throughout the genome. P1 adaptors with unique barcodes were ligated to the digested DNA of each individual fish within each population so that libraries for each fish could be pooled and sequenced together and sequences from each fish could be bioinformatically recovered. Individual samples were then pooled and sheared. Sheared fragments of 300–500 bp in length were size selected and a second adapter, P2, was ligated to the fragments. Primers specific to sequences located on the P1 and P2 adaptors were used to perform a PCR reaction to enrich only for fragments with both a P1 and P2 adaptor in 12 cycles of amplification. The resulting amplified library was cleaned with 0.6x Omega Mag-Bind Total Pure beads to remove DNA fragments <100 bp in length. The resulting library was then sequenced using a HiSeq4000 (Illumina, San Diego, CA, USA) at the University of Oregon Genomics Core Facility. 

Stacks software was utilized for processing and analyzing the RAD-seq data. We de-multiplexed the libraries by barcode and filtered RAD reads with <90% probability of being correct (based on phred score) using the “process_radtags” function in the Stacks software pipeline [38]. Processed RAD tags were then aligned against the stickleback genome using GSNAP [39]. Aligned tags were then run through the Stacks pipeline using “rxstacks” to identify and call single-nucleotide polymorphism (SNP)s throughout the genome and create a catalog of SNPs from all populations included. Genomic reads that aligned 100% to stickleback mitochondrial DNA were removed using bowtie/2.2.9 [40]. 

Genome-wide estimates of divergence (F_ST_) were calculated for each SNP and then averaged using the populations program within the Stacks software framework [38]. To reduce the data to a computationally manageable size, 1000 randomly sampled SNPs were generated using a Perl script as described in the Stacks. These were further reduced to 391 after filtering.

### 2.4. Stickleback Population Structure

In order to confirm and quantify the previously observed genetic structure and substructure of the six Oregon stickleback populations, we used principal component analyses (PCA) in R version 3.4.2 [41] with the subset of 391 SNPs. This dataset included RAD-seq data generated from three populations that were previously analyzed [30,37], and three for which collections were made and new data generated in the present study (Dean Creek, Lily Lake, and Eel Creek).

### 2.5. Microbiome DNA Extraction and 16S Sequencing

Frozen guts were weighed, rapidly thawed, and samples were immediately processed for DNA extraction using a modified Dneasy Blood and Tissue Kit protocol [42]. Guts were transferred to preheated Qiagen PowerBead Solution in QIAmp PowerFecal DNA Kit garnet bead lysis tubes and homogenized on a FastPrep120. A volume of each homogenate equivalent to 20 mg of frozen gut was transferred to preheated Qiagen ATL Buffer in 1.5 mL RINO screw-cap tubes containing 100 μL ZrOB015 beads (Next Advance, Troy, NY, USA) and homogenized again. Tissues were digested using proteinase K and incubating at 56 °C for 30 min. Four microliters of Rnase A were added, and samples were incubated at 37 °C for 30 min. The remaining steps followed Dneasy Blood and Tissue Kit protocol (Qiagen #69504), except that 800 µL of AL buffer and 800 µL of 100% ETOH were used followed with passing the supernatant through the spin column in 600 µL increments. DNA was quantified using Invitrogen Qubit Broad Range Assay and stored at −20 °C until prepped for sequencing. Total yields of gDNA were ~2 ng on average consistent with other fish gut microbiome reports (sample average = 176.88 ng/µL gDNA).

Gut microbial DNA libraries were fully processed and sequenced in the Genomics and Cell Characterization Core Facility (GC3F) at the University of Oregon. PCR was performed with barcoded V4 bacterial 16S ribosomal RNA (rRNA) primers (515F GTGCCAGCMGCCGCGGTAA, 806R GGACTACHVGGGTWTCTAAT) with an annealing temperature of 61 °C using Q5 Hot Start HiFi PCR master mix (NEB, Ipswich, MA, USA) for 27 cycles. Abundant genomic template and primers that remained in the PCR product after PCR were reduced using standard magnetic bead PCR cleanup methods. Sequencing and adonis multiplexing followed default Illumina paired-end 150 library protocols (Illumina HiSeq4000). 

### 2.6. Microbiome DNA Sequence Processing Using DADA2 Pipeline

Illumina sequencing from two independent processing times (2015 and 2017) resulted in a total of 38,290,364 high quality raw sequence reads. These reads were imported into R version 3.4.2 and processed using DADA2 version 1.6 [43] with default parameters except paired-end 150 bp reads were trimmed (*f* = 145 and *r* = 140). This resulted in 21,962,537 non-chimeric merged reads, or 198,835 reads per gut on average (process reads detailed in Table 2). In general, this process involved trimming, denoising, merging, truncating reads (reads between 250 bp and 257 bp were kept), and removing chimeras as described in DADA2 package literature [43]. We constructed a table of 100% unique 16S ribosomal DNA amplicon sequences variants (ASVs) from these processed reads and assigned taxonomy to the resulting 26,506 ASVs using the Ribosomal Database Project classifier [44].

We removed sequences from our ASV table that classified either as Archaea or chloroplast DNA, as well as sequences found in our negative controls (8097 and 688 of 26,506 ASVs, respectively). Using supervised filtering we also removed the Synergistetes and Chlorobi phyla that we suspected might be the result of sequencing error based on their extremely poor representation in our dataset. ASVs without taxonomic assignment were also filtered out. We lastly executed prevalence threshold filtering at a level of 1% of the total samples. We made a random tree using the “rtree” function in the ape package [45] in R. A phyloseq dataset was made using the “phlyoseq” function in the phyloseq R package v1.22.3 [46]. We merged the ASV table, sample data, and rtree into a phyloseq dataset object for downstream analyses.

### 2.7. Microbiome Diversity 

We found that variance grew with mean ASV abundance across gut samples, and thus we corrected this lack of homoscedasticity by square root transforming our ASV abundance data. We were concerned that differences in the number of samples and low sequencing depth in a few of the samples would influence diversity measurements. To correct any potential related errors, samples with <1000 reads were removed from the dataset (which resulted in the loss of one sample from the dataset) and then sixteen samples from each population were randomly selected without replacement to generate a new dataset (resulted in a subset of 16,397 ASVs). Random resampling at a variety of depths (15–100 samples) did not change the interpretation of linear mixed model (LMM) statistical testing results, allowing us to select a reduced subset of samples in order to improve the efficiency of our computational analyses downstream. Using this subset of these data, we assessed gut microbiome alpha diversity in terms of Simpson and Shannon diversity indices using the diversity function in the vegan v2.4.2 R package [47]. We transformed the Shannon index (1/D) to calculate the effective species number. Using the default rtree we created in phyloseq, beta diversity was calculated using both weighted and unweighted UniFrac phylogenetic distances among fish gut microbiome using the distance function in the phyloseq v1.22.3 [45] R package. These matrices were transformed into distance to centroid values for each gut community using the “betadisper” function in the vegan v2.4.2 [47] R package.

### 2.8. Linear Mixed Models

In order to determine the relative ability of genetic divergence (F_ST_), environment (estuary or fresh water), and sample period (time) to predict gut bacterial community diversity, we chose LMM approach. This method was selected in order to integrate predictive variables that are both binary and continuous into a single model. We included the sample period in order to account for potential effects caused by collecting samples at different times; in this case, two years apart. To test how relatively well these factors predicted gut ASV diversity among fish populations, we used inverse Simpson and effective Shannon diversities (alpha), as well as both weighted and unweighted UniFrac distance-to-centroid metrics (beta), as response variables in this LMM. 

We tested how well variables predicted gut microbiome diversity metrics using the following linear mixed model:Diversity = Environment + Sample Period + Population F_ST,_(1) where “environment” is a fixed term (freshwater or estuary), “sample period” is also a fixed term (2015 or 2017), and “population F_ST_” (population average genetic distance) is a random variable.

Models were fit using restricted maximum likelihood methodology using the “lmer” function in the lme4 package [48] in R. Using the step function in the lmerTest [49] R package, non-significant effects in LMMs were eliminated in a backward manner starting with random predictor variables, followed by fixed variables. Population and difference of means were calculated for the fixed part of the model and a final model provided. For hypothesis tests, the *p*-values for the fixed effect (estuary or freshwater) were calculated from an F test based on Sattethwaite’s approximation, rather than the pooled standard error formula, in order to deal with differences in standard deviation among samples. The *p*-values for random effects (population genetic structure, sample period, and starting DNA concentration) in our models were based on likelihood ratio tests (*χ*2). Distance among sites in terms of both river miles and land were tested for collinearity with host genetic distance using a Pearson’s product-moment correlation. Distance was strongly colinear with population distance, thus distance was not included in LMMs to avoid inflation of model estimates.

### 2.9. Microbiome Composition and Permutational Multivariate Analysis of Variance (PERMANOVA)

We transformed ASV counts to relative abundances in order to reduce the strong effects of potentially over-sampled taxa. Principal Coordinate Analysis (PCoA) ordination was performed with these relative abundances and square root transformed Bray–Curtis dissimilarity metrics using the function ordinate in R package phyloseq v1.22.3 [46]. We did this to determine if ASVs covaried among fish gut communities in order to choose the appropriate statistical approach. A permutational multivariate analysis of variance was used instead of a linear mixed model approach. We made this decision based on a lack of collinearity structure among ASVs observed along any given axis plotted in multivariate spatial models (Appendix A). We included fish population average genetic distance (F_ST_), geographic distance (river miles), and population genetic heterozygosity, as well as sample period (2015 or 2017) and environment (estuary or freshwater), as fixed binary variables in our full model. We tested the relative degree to which these variables explained gut microbiome composition in terms of Bray–Curtis dissimilarity among fish guts using the “adonis” function in vegan v2.4.2 R package [47]. 

### 2.10. Amplicon Sequences Variants Abundance Enrichment Analysis

In order to determine how gut microbiome compositional variation among threespine stickleback in Oregon was influenced by host population genetic structure and environment, we first looked broadly at what bacterial groups were present in our gut samples. Successively, we quantified AVS abundance differentiation across the major geographic partitions. We did this by converting the phyloseq to a DESeq dataset using the phyloseq_to_deseq2 function in the DESeq2 v1.18.1 R package [50]. The function “estimateSizeFactors” does not handle ASV count values = 0. In order to handle zero values, we applied: gm_mean = function (x, na.rm = TRUE)^Σ(log(x[x > 0])^, na.rm = na.rm/length(x), where x is ASV counts. The “estimateDispersions” function was used to estimate dispersions. We tested gut ASVs for significant differential abundance both between inland and coastal freshwater populations and between estuary and freshwater populations using the DESeq function with the default Benjamin–Hochberg multiple-inference correction and with the fitType parameter set to ‘local’. Only ASV differential abundances that were statistically significant (*p* ≤ 0.05) were considered in our biological interpretation of these results. 

## 3. Results

### 3.1. Genetic Variation in Oregon Stickleback Is Partitioned Between Geographical Regions and Environment 

Using PCA, we found genetic variation in the populations that we sampled was mainly partitioned between geographic regions and to a lesser extent between environments, confirming previous findings [30,37]. The first PC (PC1), accounted for 46.33% of the overall genetic variation partitioned genetic variation between populations collected along the coast and those collected inland in the Willamette Basin. PC2 accounted for 8.56% of overall genetic variation grouped coastal populations separately by environment (estuary vs. freshwater) (Figure 2). In congruence with these PCA results, average population genetic divergence, as measured by F_ST_ in pairwise comparisons between coastal and inland fish populations, was on average two-fold greater (~0.242) than between inland (~0.125) and four-fold greater than that among coastal populations (~0.063) (Table 3). 

### 3.2. Stickleback Gut Microbiome Diversity and Composition Is Better Predicted by Population Genetic Divergence Than by Environmental and Geographic Differences

We found that gut communities varied in terms of both alpha and beta diversity among individual fish, as well among populations on average (Figure 3 and Figure 4). Coastal populations tended to have greater inter-individual and lower individual gut community diversity on average. 

Results of the linear model showed that gut microbiome alpha diversity was better predicted by population F_ST_ (1/Simpson: *χ*2 = 5.53, *df* = 1, *p* < 0.02 and Shannon: *χ*2 = 13.77, *df* = 1, *p* < 0.001), than by environment (1/Simpson: *F* = 2.03, *df* = 1, *p* = 0.23 and Shannon: *χ*2 = 2.02, *df* = 1, *p* = 0.22) and sample period (1/Simpson: *χ*2 = 0.0, *df* = 1, *p* = 1.0) (Figure 3 and Table 4). In addition, we found that gut microbiome beta diversity in terms of UniFrac distance was also better predicted by population F_ST_ (*χ*2 = 29.32, *df* = 1, *p* < 0.001) than by environment (*F* = 3.36, *df* = 2, *p* = 0.14) and sample period (*χ*2 = 1.0, *df* = 1, *p* = 1.0). In addition, we found that environment (*F* = 6.94, *df* = 1, *p* < 0.08) better predicted beta diversity in terms of Weighted UniFrac distance to centroid than F_ST_ (*χ*2 = 1.07, *df* = 1, *p* = 0.30) and sample period (*χ*2 = 0.22, *df* = 1, *p* = 0.64) (Figure 4 and Table 4). Lastly, we found a positive correlation between population genetic heterozygosity and beta diversity both in terms of UniFrac distance to centroid (Pearson’s product-moment correlation = 0.61, *t* =7.42, *df* = 94, *p* < 5.11 × 10^−11^) and weighted UniFrac distance to centroid (Pearson’s product-moment correlation = 0.31, *t* =3.11, *df* = 94, *p* < 0.003). The random effects of sample period were completely negligible for all diversity metrics we tested (Table 4). 

Permutational multivariate analysis of variance (PERMANOVA) showed that population genetic divergence (F_ST_: *R^2^* = 0.080, *p* < 0.001) explained the most variation relative to the other factors (Table 5). Furthermore, sample period (*R^2^* = 0.053, *p* < 0.001), population genetic heterozygosity (*R^2^* = 0.050, *p* < 0.001), environment (*R^2^* = 0.058, *p* < 0.001), and river mile (*R^2^* = 0.053, *p* < 0.001) explained similar portions of variation in gut microbiome in terms of Bray–Curtis dissimilarity among fish gut communities. F_ST_ strongly covaried with both geographic distance (Pearson’s product-moment correlation = 0.97, *t* = 39.31, *df* = 94, *p* < 1 × 10^−16^) and river miles (Pearson’s product-moment correlation = 0.94, *t* = 26.0, *df* = 94, *p* < 1 × 10^−16^), as well as population heterozygosity (Pearson’s product-moment correlation = 0.97, *t* = 41.0, *df* = 94, *p* < 2.2 × 10^−16^). The remaining majority of gut microbiome variation among these six wild fish populations remained unexplained by the factors we accounted for in this study (~71%).

### 3.3. Differential Amplicon Sequences Variants Abundance Among Gut 

We found that the stickleback microbiome in Oregon consisted of predominately bacteria in the phyla Proteobacteria, Fermicutes, Bacteroides, Actinobacteria, Acidobacteria, Planctomycetes, Verrucomicrobia, Chloroflexi, Tenericutes, and Fusobacteria (ordered by most to least abundant) based on relative abundance in the metacommunity (Appendix A). These groups are found in the gut in other wild fish surveys [36,51,52,53].

Looking more finely in a subset of 1100 of the total 16,530 bacterial ASVs and 16 of the total 23 phyla, we found a significant difference in abundance between regions of ASVs (Figure 5a). Nearly half of these ASVs (46%) comprised genera in the phylum Proteobacteria. Single ASVs were exceptionally enriched (log_2_fold > 20) in fish guts in the Proteobacteria (inland) and on the Fermicutes (coast). Nine phyla were enriched in fish guts in both inland and coastal populations. In contrast, Tenericutes were enriched only in inland freshwater fish, and Fusobacteria, Spirochaetes, Nitrospirae, Ignavibacteriae, Lentisphaerae, and Cloacimonetes were enriched only in coastal freshwater fish (Figure 5a). The Tenericutes in inland fish guts comprised one ASV where sequence most closely matched (95%) a human pathogen species, *Mycoplasma penetrans* [51,54,55]. Each phylum enriched exclusively in coastal freshwater fish guts was represented by a single ASV as well, the exception being the Spirochaetes, which was represented by two ASVs. A couple of these match most closely with bacteria species associated with host pathogenicity (*Fusobacterium varium* and *Treponema* spps.), but the rest were taxa typically associated with lake and sludge communities.

Comparing differential abundance between environments, we found significant differences in a subset of 404 ASVs comprising 11 phyla (Figure 5b). More than half (53%) of these enrichments comprised genera in the phylum Proteobacteria. There were eight ASVs that were exceptionally enriched (log_2_fold > 20) in estuary fish (one Spirochaetes) and freshwater (five Proteobacteria, one Bacteroides, and one Firmicutes). Seven phyla were enriched in gut microbiomes in both environments. In contrast, the Spirochaetes were enriched in only estuary fish, and the Fusobacteria, Gemmatimonadetes, and Verrucomicrobia were enriched in only freshwater fish (Figure 5b). The Spirochaetes enrichment in estuary fish comprised a single ASV that did not closely match any known species but classified to the Brevinema, a genus which consists of bacteria detected in fish guts in other studies [56,57]. Similar to what we found in inland freshwater fish, the phyla that were enriched in only freshwater fish comprised two ASVs that matched most closely with bacteria species associated with host pathogenicity (*Fusobacterium varium* and *Treponema* spps.), and the remainder were not well matched with bacteria that had known origins.

## 4. Discussion

### 4.1. Host Population Genomic Divergence Correlated with Gut Microbiome Divergence

We have come to appreciate the importance of the host-associated microbiome in vertebrate health and disease. This current understanding is due to recent rapid advances in this field, primarily using experimental vertebrate models (e.g., mice and fish). However, despite our awareness and efforts, we still lack a complete understanding of the relative roles of environment and host genetic variation in host microbiome assembly. Studies of microbiomes of hosts in the wild can help fill this knowledge gap. Wild hosts can possess incredible genetic variation and are found among diverse environments, and we often know much about their natural evolution and ecology, yet little work in this field has taken advantage of this. In addition, studies that have been done using wild populations have not broadly considered the host genome. Rather, they have focused narrowly on specific genes or small regions of the host genome deemed important by experimental work. This is an important issue to address in order to understand how experimental research should be interpreted and to advance the field.

Here, we addressed this problem by collecting both host population genomic and microbiome data from wild populations of threespine stickleback fish that inhabit various environments in Oregon. We hypothesized that if host genetics played an important role in gut microbiome assembly, then, with a robust level of population genetic resolution, we would find a pattern in the gut metacommunity that reflected the genetic relationship among populations of wild hosts. We selected six populations of stickleback situated in Oregon among three watersheds over a range of geographic distances (~12–80 km apart as a crow flies) and across two distinct environments (estuary and freshwater). Doing so allowed us to measure the relative degree to which each factor explained variation in gut microbiome among hosts. 

Using several hundred single nucleotide polymorphisms randomly spread throughout the genome, we first confirmed a strong genetic differentiation between inland and coastal populations, as previously seen in Oregon [30,37]. In addition, on the coast, estuary populations further differentiated, although to less an extent, from freshwater populations. Using metabarcoding data (16s rRNA) from the gut communities of these fish, we found that gut microbiome varied among both individual fish and fish populations in terms of composition and diversity in threespine stickleback in Oregon. We were, therefore, able to produce corresponding host genetic and microbiome data from natural populations of a vertebrate that is also amenable to manipulative microbiome studies in the laboratory [29]. In contrast to most model vertebrate organisms, in stickleback we can now integrate a much better understanding of the relative patterning of host genomic and microbiome diversity in the wild with the power of subsequent manipulative studies.

Our results provide evidence that genome-wide genetic variation among host populations can influence divergence in the gut microbiome. Our findings agree with findings in other systems that relatively more similar gut microbiomes exist among related as compared to unrelated individuals [10,20,21,23,58] and that aspects of the gut microbiome can act as complex and heritable host traits [24,25,26]. Our findings also align with previous work in Canadian threespine stickleback, in which they found that average gut microbiome phylogenetic distance and host population genetic divergence in six satellite markers positively correlated [36]. Our work expands our understanding of both the geographic and genetic scales at which this pattern is evident in wild threespine stickleback. We found similar results among populations in Oregon separated by ~12–80 km. 

We found that both alpha and beta diversity varied among wild stickleback fish populations in Oregon. This variation in gut diversity was primarily predicted by population genetic divergence among these populations. In addition, we found that population genetic divergence better explained a partition in gut microbiome composition in Oregon than environment, geography, sample period, and host population genetic heterozygosity. This evidence supports previous findings that inner-host genetics have a more important role in the assembly of the gut-associated microbiome in wild hosts than geography and environment. In previous work conducted on Canadian stickleback populations, gut alpha diversity was negatively correlated with heterozygosity in Major Histocompatibility Complex class II (MHCII) alleles [35]. In congruence with this finding from Canada, Oregon populations demonstrated a weak negative correlation between average genome-wide heterozygosity and gut alpha diversity (*t* = −1.23, *df* = 94, *p* = 0.22). It was also previously suggested that if high allelic heterozygosity in *MHCII* genes was a proxy for high host genetic heterozygosity, then higher host population heterozygosity may have explained lower beta diversity in Canadian stickleback populations [35]. In contrast, we found that heterozygosity among stickleback populations in Oregon positively correlated with gut microbiome beta diversity. Therefore, it is unclear whether genetic heterozygosity influences gut microbiome diversity. We observed that Oregon coastal stickleback populations had higher genetic heterozygosity than inland populations, likely due to higher rates of gene flow with diverse marine populations along the coast. We speculate that this mechanism could in part influence the greater inter-individual variation in the gut microbiome (beta-diversity) in the coastal populations we observed, especially if this genetic variation occurs in traits important in gut assembly. Lastly, random sampling of microbes from the source pool and the stochastic loss and replacement of microbes can also explain variation in gut microbiomes [12]. These “neutral processes” in gut microbiome assembly have not yet been studied extensively in wild populations and have never been explored in threespine stickleback. We acknowledge these caveats, and thus our conclusions should be taken as initial results that warrant subsequent studies. 

Even though host population genetic divergence best predicted gut diversity, environment also explained a portion of the overall variation in the gut microbiome. This was corroborated by our finding that environment best predicted gut beta diversity (weighted UniFrac distance). In addition, stickleback populations found in estuaries had on average higher beta diversity and lower alpha diversity than freshwater populations. This greater inter-individual variation in estuary fish could have been due to the exposure of estuarian fish to a more diverse pool of microbes. Stickleback acquire their microbes in part through diet, sediment, and water sources [33,36]. These inputs all vary among freshwater, marine, and estuary environments. Furthermore, estuarian environments can be a unique blend of both freshwater and marine inputs both in terms of microbiota and nutrients important in microbial growth [16]. Stickleback fish can traverse and have varying residency times among these environments in Oregon. Therefore, we speculate that this diverse exposure of the host among marine, estuary, and freshwater environments might have contributed to gut microbiome diversity among individual stickleback. Future studies in wild populations could focus on microbiomes associated with host food sources, food types and nutritional content of the varying food types found among the different environments, and how these factors and those studied here shape the gut microbiota in other species that inhabit a variety of environments (e.g., salmonids). 

### 4.2. A Small Subset of Microbial Taxa Contributed to Microbiome Divergence Linked to Host Genetics and Environment

Given that population genetic divergence explained a portion of gut metacommunity variation, we wanted to know which microbial taxa contributed to significant differential abundance between inland and coastal freshwater fish. We found that a small subset of the gut metacommunity was significantly differentiated in terms of abundance across the main divergence in host population genetic structure in Oregon. These ASVs represent a group of bacteria that could strongly interact with the host in terms of genome-wide allele variation by being heritable and important in host fitness. This notion seemed plausible considering the history of threespine stickleback. These ASVs, if isolated, could be used to address the question of differential colonization in lab fish raised free of microbes, a benefit of work in a system that exhibits both natural variation and that can be manipulated in the laboratory. 

It is unclear if host-environment interactions can influence the gut microbiome indirectly via the host. There was a subset of phyla comprised of ASVs that were significantly enriched in both or only one of these environments. A smaller subset of these was enriched log_2_fold > 20 suggesting a strong enrichment due to environment-related processes. We speculate that a strong association of some taxa in the gut can be specific to the environments in which hosts are found. This specificity could be due to the sorting of the microbial pool in the environment. However, it has been reported that microbes in the environment do not predict what is found in the stickleback gut [36]. Further testing in stickleback could help illuminate these aspects of gut assembly. For instance, parallel work in lab experimentation and using bacterial strains curated from the wild incorporated with environmental gradients could address questions of whether host–environment–microbe interactions can occur. 

We also observed that estuary and freshwater fish had different gut communities in terms of the relative abundance of major phyla present in the gut, as was the case in another study in Canada [36]. The Firmicutes in estuary fish and the Proteobacteria in freshwater fish were the most prevalent phyla. Differences in prevalent taxa found in the gut microbiome among environments could in part be due to variation in factors that differentiate the microbial pool available among environments. Variation in the microbial pool across space can lead to differential colonization among hosts occupying different environments. For example, salinity is an important variable that globally sorts microbial communities in the environment [59].

### 4.3. Study Limitations

Using comparative studies to document patterns of microbiome variation in natural populations of organisms that vary in terms of host genetics and environments is very important in providing foundational knowledge of potential mechanisms affecting host-microbe relationships in the wild. However, we acknowledge that the design of this study is limited by the periodicity of the collections and that the observed effect of genetic divergence could be explained by the combination of environment, sampling period, and geographical location. Therefore, as is the case with many comparative studies, these results are preliminary and further validation is needed through manipulative experiments. Such studies should include an expansion of populations collected during the same time frame and studies that bring wild populations into the lab and grown under common conditions to reduce the influence of environment. 

## 5. Conclusions

We studied stickleback gut microbiomes in the wild and found that host population genetic divergence (based on 391 randomly selected genome-wide SNPs) explained gut microbiome partitioning, in terms of both diversity and composition, better than geography, environment, sample period, and population genetic heterozygosity. In addition, geography, environment, sample period, and population genetic heterozygosity equally influenced the gut microbiome. Lastly, we found that relatively small subsets of the gut microbiome in wild stickleback in Oregon uniquely differentiated in terms abundance across different factors. 

## Figures and Tables

**Figure 1 genes-10-00484-f001:**
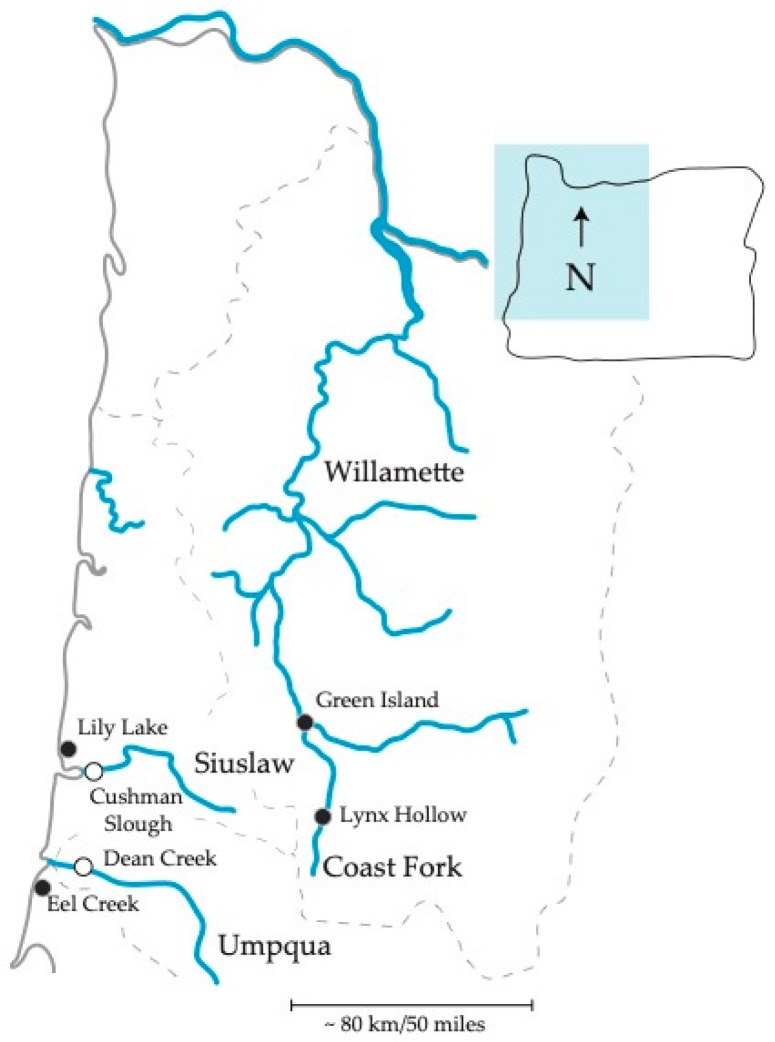
Site map in Oregon. Filled and open points represent freshwater and estuary, respectively. Two inland freshwater populations and four coastal populations were sampled.

**Figure 2 genes-10-00484-f002:**
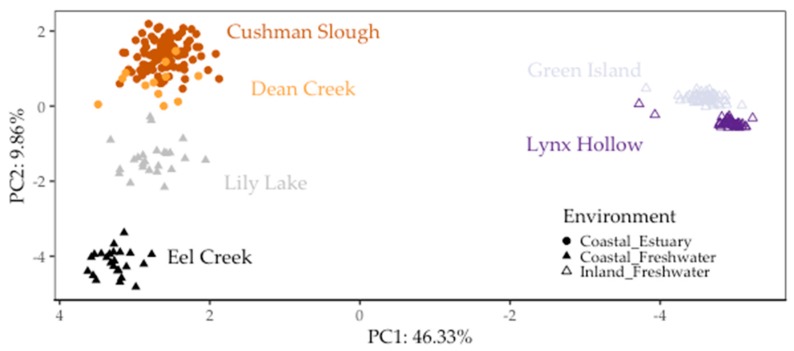
Threespine stickleback population genetic structure partitioned between inland and coastal sites, and between estuary and freshwater sites on the coast. This biplot includes the first two principal component (PC) axes, which accounted for more than half (56.19%) of the overall variation in 391 randomly selected genome-wide SNPs for the six populations in this study [37]. Each point represents a fish. Shape fill represents whether sites were in the Willamette Basin (“Inland”) or in watershed along the coast (“Coastal”). Colors represent collection sites (“populations”) in Oregon. Populations in Oregon based on restriction site-associated DNA sequencing (RAD-seq) single nucleotide polymorphisms (SNPs).

**Figure 3 genes-10-00484-f003:**
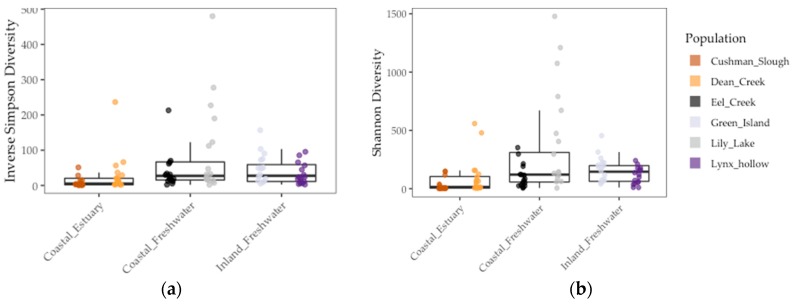
Gut microbiome alpha diversity among threespine stickleback populations in Oregon in terms of inverse Simpson diversity (**a**) and Shannon diversity (**b**) measures. Colors represent collection sites (“populations”). Each point is a fish gut. Mid-box lines are pooled means of major environments (e.g., Coastal Freshwater”). Box whiskers are pooled standard deviation of major environments, as well.

**Figure 4 genes-10-00484-f004:**
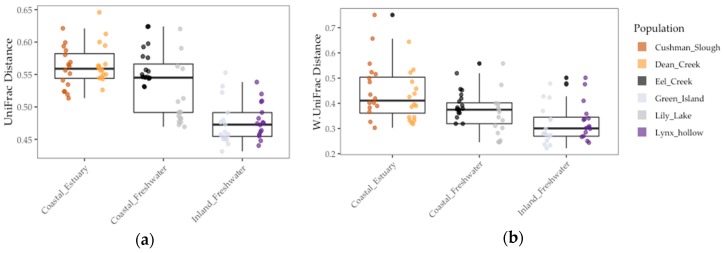
Gut microbiome beta diversity among threespine stickleback populations in terms of distance-to-centroid transformed: (**a**) Unweighted UniFrac distance (**b**) Weighted UniFrac distance. Colors represent collection sites (“populations”). Each point is a fish gut. Mid-box lines are pooled means of major environments (e.g., Coastal Freshwater”). Box whiskers are pooled standard deviation of major environments, as well.

**Figure 5 genes-10-00484-f005:**
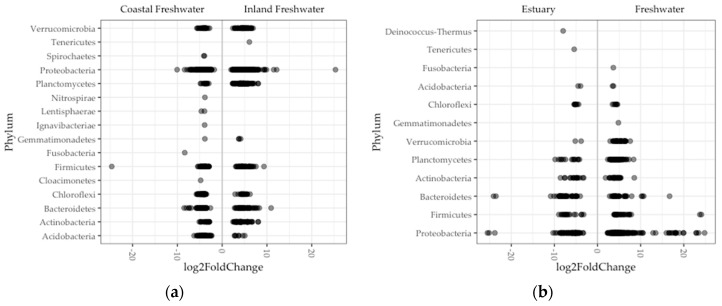
Significant differential ASV abundance in fish gut microbiome was determined between: (**a**) Inland and coastal regions; (**b**) and freshwater and estuary environments in Oregon. Each point (darker points are overlapped points) is an ASV that is enriched in a phylum. Negative and positive log_2_fold values are enrichments in on either side of x-axis, for example, in estuary and freshwater fish guts, respectively.

**Table 1 genes-10-00484-t001:** Sample collection details for sites along the coast and inland in Oregon.

Site	Region	Fin Clip Collected	Environment	Location	Gut Collected	Guts
Dean Creek	Coast	2017 [37]	Estuary	N43.6960, W124.0170	06/13/17	20
Eel Creek	Coast	2017 [37]	Freshwater	N43.5869, W124.1861	06/13/17	20
Cushman Slough	Coast	2007,2009 [30]	Estuary	N43.9881, W124.0395	07/29/15	22
Lily Lake	Coast	2015 [37]	Freshwater	N44.0886, W124.1140	08/04/15	22
Green Island	Inland	2013 [37]	Freshwater	N44.1582, W123.1189	07/24/15	22
Lynx Hollow	Inland	2013 [37]	Freshwater	N43.8613, W123.0249	07/03/17	20

**Table 2 genes-10-00484-t002:** Stepwise number of reads in each population after each step during DADA2 processing of 16S bacterial sequence reads.

Site	Input	Filtered	Merged	Nonchimeric
Cushman Slough	9,256,421	4,888,393	4,298,407	4,264,096
Dean Creek	3,359,145	3,069,584	2,521,985	2,204,735
Eel Creek	2,287,323	2,055,175	1,905,651	1,663,629
Lily Lake	10,541,264	7,072,180	6,777,026	6,721,944
Green Island	8,616,378	4,260,381	3,699,754	3,681,706
Lynx Hollow	4,232,833	3,939,422	3,624,234	3,426,427

**Table 3 genes-10-00484-t003:** Population genetic divergence (F_ST_) among six threespine stickleback populations in Oregon based on 391 RAD-seq SNPs. The populations program in Stacks software was used for this output.

	Eel Creek	Cushman Slough	Lily Lake	Green Island	Lynx Hollow
Dean Creek	0.1293	0.0212	0.0698	0.3117	0.3942
Eel Creek		0.0919	0.0793	0.4709	0.5748
Cushman Slough			0.0653	0.2392	0.2789
Lily Lake				0.3028	0.3661
Green Island					0.1488

**Table 4 genes-10-00484-t004:** Linear mixed model (LMM) statistics results. Results with *p*-value ≥ 0.05 in bold. Guts rarefied to even depth of 1000 amplicon sequences variants (ASVs) per sample.

LMM	Inverse Simpson			Shannon			Unweighted UniFrac			Weighted UniFrac		
**Fixed**	***F***	***df***	***p***	***F***	***df***	***p***	***F***	***df***	***p***	***F***	***df***	***p***
Environment	2.03	1	0.23	2.02	1	0.22	3.36	1	0.14	6.94	1	<0.08
**Random**	***χ**2***			***χ**2***			***χ**2***			***χ**2***		
Pop. F_ST_	**5.53**	**1**	**<0.02**	**13.77**	**1**	**<0.001**	**29.32**	**1**	**<0.001**	1.07	1	0.30
Sample Period	0.0	1	1.0	0.00	1	1.0	0.0	1	1.0	0.22	1	0.64

**Table 5 genes-10-00484-t005:** PERMANOVA (in order of proportion explained) statistics results.

PERMANOVA	*R^2^*	*df*	*p-*value
Population F_ST_	0.080	1	0.001
Environment	0.058	1	0.001
Sample Period	0.053	1	0.001
River Miles	0.053	1	0.001
Population Heterozygosity	0.050	1	0.001

PERMANOVA: Permutational multivariate analysis of variance.

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
