# Peer review of "Population Genetic Divergence and Environment Influence the Gut Microbiome in Oregon Threespine Stickleback"

_genes, 2019, doi:10.3390/genes10070484_

Reviewer 1 Report

In their study, Steury et al. investigated the gut microbiome in Oregon Threespine Stickleback. The authors assessed the genetic diversity of the studied fish species by SNP analysis. In addition, the authors assessed the gut microbiome by 16S rRNA gene sequencing targeting the V4 region.

Although the manuscript is well written, I have one main concern. In my opinion, the study design is too weak to support the main results which is emphasized  by the low degree of freedom. Two samples were taken form estuaries but in different years, two samples were taken close to the coast but in different years and two samples were taken from a different freshwater system again in two different years. The authors emphasize the importance of studies in natural settings which I fully agree with. However, the observed effect of genetic diversity on bacterial community structure might be also caused be a combination of environment, sampling period and geographical location. To test the effect of genetic diversity, I would use different Stickleback populations and incubate them under different environmental conditions in a aquaculture experiment.

Please include a study limitation section in your manuscript and add a sentence that the results are preliminary and need further validation.

Minor comments

Abstract:

More results, less background

Methods:

How was the tree generated for UniFrac analysis?

Conclusion:

The second part is not needed and should be removed.

Author Response

Please include a study limitation section in your manuscript and add a sentence that the results are preliminary and need further validation.

We have added a section to the end of the paper entitled ‘4.3 Study Limitations’ (Page 14 lines 537-546). We agree with the reviewer that inferences about causal mechanisms should not be extended too far, but think that documentation of patterns in natural populations that generate preliminary conclusions and testable hypotheses is a worthwhile goal. We feel that by adding this section, and some additional changes throughout the Discussion, we have sufficiently added appropriate levels of caution to our conclusions without significantly diminishing the impact of our findings.

Reviewer: “Abstract: More results, less background”

We appreciate the reviewers comments and have made some small changes to the abstract to reflect. Page 1 lines 11-23

Reviewer: “Methods: How was the tree generated for UniFrac analysis?”

This is now described in the methods. Page 6 line 224

Reviewer: Conclusion: “The second part is not needed and should be removed.”

Removed. Page 14

Reviewer 2 Report

The document was well written and supports previous regarding wild vertebrate genetics, environment and their microbiome community structure. Such information is relatively scarce compared to laboratory research and as such makes a significant contribution to the field. 

Data supports authors' conclusions. Methods are well described and are appropriate and current.  

Author Response

Thank you!

Reviewer 3 Report

The MS by Steury et al. aims to relate diversity and community structure of gut microbiome in the Oregon Threespine Stickelback and fish population genomic. Although the idea is not fully original as other previous studies related population genetics of this specie with host gut microbiome, this is the first time a wide population genomic approach is taken. The introduction is well written and analysis are appropriate and correctly applied, however, there is a lack of fluidity and structure in the way the paper is written. The M&M section should include all analysis steps undertaken and be accompanied by explanation of why specific analyses were chosen over others. The Results should briefly state the outcomes without repeating information and reasons for choosing such analysis. The discussion is in some part too speculative and should discuss more critically real drawbacks of this study and potential improvements. In light of the above I recommend publication of the MS after careful revision, at this regards I have some suggestion for the authors:

Line 44, add some references

Lines 84-86, this statement is not clear I suggest a revision, interindividual variation is here stressed but I do not find similar statement in the rest of the paper or even in the discussion, or at least not emphasised as here.

Lines 197-198, not clear to me why 16 out of 20 samples were resampled in each population and how this was made

Line 210-211, is this a repetition with what stated in Line 197-198? Authors should clarify these steps

Line 223 – here authors should explain before everything else, the aims of the analysis and why they choose LMM

Line 223, gut microbiome diversity

Lines 242-243 no need to add this information, AVS relative abundance (out of the total reads per host individual sample) is already enough clear and short statement

Line 247-248, this sentence is not very clear should be rewritten

Line 256, I think the readers would appreciated to read first the reason to apply such analysis (that is discover late in results, but the way the paper is structured is confusing) A statement like the follow could help: “ In order to determine how gut microbiome compositional variation among threespine …. was influenced by host population genetic structure and environments, we first looked broadly at what bacterial group were present in our samples. Successively we quantify AVS abundance differentiation across major geographic partition in host population structure. We did this by …………”

also see comments below, It is necessary to transfer some details on methodologies listed in the result section to the M&M section (in this case I move sentences in lines 398-400, and  lines 406-408 to line 256.

Line 267, 3.1-paragraph title: Stickleback population genetic structure partitioning

Lines 271 277, all this part should be moved to M&M, to my opinion a separate M&M paragraph on fish population structure should be added

Line 271, authors stated that they selected 6 populations while successively declaring that the dataset included data generated by 4 previously analysed populations and 3 for which new collections were made….but in this way the number of populations amount to 7, authors should explain or correct this number.

Line 272, Table 1 not Table 2

Line 273, authors should state clearly how PCA was performed and the variables used, in M&M not in results

Line 300, title paragr. 3.2, unless requested by the journal I would short the paragraph title as “Stickleback gut microbiome diversity and composition”  

Lines 303-304, no need to add this statement

Line 313, is taxa abundance a typing mistake? is this a rarefaction step as the one mentioned in line 209? Not clear, please clarify

Lines 317-323, this paragraph should be moved to M&M and included in the par 2.7

Line 328, how rarefaction on AVS counts was made? Not clear to me, do you here refer to samples rarefaction at 1000 reads/sample?

Line 331, the statement should be briefly introduced….”Results of the Linear Model …..showed that gut microbiome alpha diversity was ……”

Lines 351-354, this paragraph should be moved to M&M par 2.8,

Lines 354-365 similarly, authors should be consistent in reporting all methodological choices and reasons for such choices in the M&M section together with details on methodology and provide in results only outcomes of the analyses.

Line 372, introduce the analysis briefly to make it easier for the reader to follow the flow of the paper, “ Results of permanova analyses showed that population genetic divergence …”

Line 3.3 similar comments as for paragraph 3.2

Line 398-400, and lines 406-409, authors should once again be consistent in reporting all methodological choices and reasons for such choices in the M&M section together with details on methodology and provide in results only outcomes of analyses.

Line 483-484-need to add some references

Line 489-492, previous work done in ref 36 should be a bit more emphasized in the introduction section, justifying the need for the present study in the context of a deeper genomic analysis.

Line 500, confusing statement; the adjectives “reduced” and “high” should be put in brackets or deleted

Line 518-535 in this section authors should discuss issues such as the need to consider the influence of microbiome associated to host food sources, way of nutrition and diets typology for future studies on wild population of other host microbiome systems.

Line 540-560, Par 4.2 is very speculative and can be reduced drastically, focusing more on the results of this study, or on new questions raised from this study…..just to give an example could be interesting to discuss the unexplained variance ( which is quite high, 70%), and guess potential sources of variation based on literature.

Author Response

In light of the above I recommend publication of the MS after careful revision, at this regards I have some suggestion for the authors:

Reviewer: “Line 44, add some references”

We have changed the wording leading up to the references to better reflect our statement. We feel that the references provided reflect this change. Page 1 lines 43

Reviewer: “Lines 84-86, this statement is not clear I suggest a revision, interindividual variation is here stressed but I do not find similar statement in the rest of the paper or even in the discussion, or at least not emphasised as here.”

We have changed this statement to reflect the major findings of this research. Page 2 lines 90-92

Reviewer: “Lines 197-198, not clear to me why 16 out of 20 samples were resampled in each population and how this was made

The manuscript has been changed to address this. Page 6 lines 216-221

Reviewer: “Line 210-211, is this a repetition with what stated in Line 197-198? Authors should clarify these steps”

This has been clarified. Page 6 lines 212-228

Reviewer: “Line 223 – here authors should explain before everything else, the aims of the analysis and why they choose LMM”

A better introduction to this section has been added. Page 6 lines 232-235

Reviewer: “Line 223, gut microbiome diversity”

We assume this should have been line 225 and was fixed. Page 7 lines 240

Reviewer: “Lines 242-243 no need to add this information, AVS relative abundance (out of the total reads per host individual sample) is already enough clear and short statement”

This was removed. Page 7 line 259 

Reviewer: “Line 247-248, this sentence is not very clear should be rewritten”

This section was re-written to clarify our intent. Page 7 lines 260-263

Reviewer: “Line 256, I think the readers would appreciated to read first the reason to apply such analysis (that is discover late in results, but the way the paper is structured is confusing) A statement like the follow could help: “ In order to determine how gut microbiome compositional variation among threespine …. was influenced by host population genetic structure and environments, we first looked broadly at what bacterial group were present in our samples. Successively we quantify AVS abundance differentiation across major geographic partition in host population structure. We did this by …………” also see comments below, It is necessary to transfer some details on methodologies listed in the result section to the M&M section (in this case I move sentences in lines 398-400, and  lines 406-408 to line 256.”

Explanation of the methodologies used was added to the methods section as the reviewer suggested. Page 7 lines 260-263

Reviewer: “Line 267, 3.1-paragraph title: Stickleback population genetic structure partitioning”

This title was changed to make it more clear. Page 8 lines 289-290

Reviewer: “Lines 271 277, all this part should be moved to M&M, to my opinion a separate M&M paragraph on fish population structure should be added”

This was moved to the methods section and a Population Structure section was added to the methods. Pages 3 & 5  lines 97-98 & 152-157

Reviewer: “Line 271, authors stated that they selected 6 populations while successively declaring that the dataset included data generated by 4 previously analysed populations and 3 for which new collections were made….but in this way the number of populations amount to 7, authors should explain or correct this number.”

This has been corrected to three previous and three additional populations. Page 5 lines 155-156

Reviewer: “Line 272, Table 1 not Table 2”

This sentence has been removed

Reviewer: “Line 273, authors should state clearly how PCA was performed and the variables used, in M&M not in results”

A new section was added to the methods and a description of how the PCA was performed and what variables were used is included. Page 5 lines 152-157

Reviewer: “Line 300, title paragr. 3.2, unless requested by the journal I would short the paragraph title as “Stickleback gut microbiome diversity and composition”. 

We feel that this an adequate title for this section as it is a statement of finding. 

Reviewer: “Lines 303-304, no need to add this statement”

Removed. Page 8 line 318

Reviewer: “Line 313, is taxa abundance a typing mistake? is this a rarefaction step as the one mentioned in line 209? Not clear, please clarify”

This has been removed. Page 9 line 325

Reviewer: “Lines 317-323, this paragraph should be moved to M&M and included in the par 2.7”

This was moved to the M&M section. Page 6&7 lines 235-239

Reviewer: “Line 328, how rarefaction on AVS counts was made? Not clear to me, do you here refer to samples rarefaction at 1000 reads/sample?”

This is described in the methods section 2.7. Page 6 lines 216 -221

Reviewer: “Line 331, the statement should be briefly introduced….”Results of the Linear Model …..showed that gut microbiome alpha diversity was ……”

An introduction was added. Page 9 lines 337

Reviewer: “Lines 351-354, this paragraph should be moved to M&M par 2.8,”

This paragraph has been removed and M&M reflect what was said here.Page 7 lines 260-263

Reviewer: “Lines 354-365 similarly, authors should be consistent in reporting all methodological choices and reasons for such choices in the M&M section together with details on methodology and provide in results only outcomes of the analyses.”

All methodological details have been moved to the M&M section.Page 7 lines 260-263

Reviewer: “ Line 372, introduce the analysis briefly to make it easier for the reader to follow the flow of the paper, “ Results of permanova analyses showed that population genetic divergence …”

An introduction has been added. Page 10 lines 358

Reviewer: “Line 3.3 similar comments as for paragraph 3.2”

Changes have been made to reflect the comments in paragraph 3.2. Page 7 lines 275-278

Reviewer: “Line 398-400, and lines 406-409, authors should once again be consistent in reporting all methodological choices and reasons for such choices in the M&M section together with details on methodology and provide in results only outcomes of analyses.”

All methodological details have been moved to the M&M section. Page 7 & 10 lines 275-278, 375, 381.

Reviewer: “Line 483-484-need to add some references”

This statement has been removed. Page 12 line 451

Reviewer: “Line 489-492, previous work done in ref 36 should be a bit more emphasized in the introduction section, justifying the need for the present study in the context of a deeper genomic analysis.”

This justification has been added to the intro. Page 2 lines 83-87

Reviewer: “Line 500, confusing statement; the adjectives “reduced” and “high” should be put in brackets or deleted”

These terms have been removed. Page 12 line 466

Reviewer: “Line 518-535 in this section authors should discuss issues such as the need to consider the influence of microbiome associated to host food sources, way of nutrition and diets typology for future studies on wild population of other host microbiome systems.”

A small section has been added that includes these suggestions. Page 13 lines 497-501

Reviewer: “Line 540-560, Par 4.2 is very speculative and can be reduced drastically, focusing more on the results of this study, or on new questions raised from this study…..just to give an example could be interesting to discuss the unexplained variance ( which is quite high, 70%), and guess potential sources of variation based on literature.”

We have written this section to be more streamlined as the reviewer suggests.

Round  2

Reviewer 3 Report

additional minor comments of revised version:

Line 17 -  genome-wide population genomic, or better wide population genomic

Line 358 PERMANOVA

Line 375 “Looking broadly at what bacterial groups were present in stickleback gut samples, we found that the stickleback microbiome in Oregon consisted of predominately bacteria in the phyla…”

Too redundant could be instead “We found that the stickleback microbiome in Oregon consisted of predominately bacteria in the phyla…”

Line 381- Looking at a finer scale between regions we found significant differential abundance in a subset  of 1,100 of the total 16,530 bacterial ASVs and 16 of the total 23 phyla we documented (Figure 5a).

This phrase is not clear

Line 500 – “that inhabit a variety of environments (e.g. many 500 salmonids)”  I would delete this and leave only “(e.g. salmonids)”

Author Response

Thank you for your final five, fine edits to the manuscript. 

We have made each of the changes that you suggested using track changes mode in the manuscript. 

Because the modifications are relatively straight forward typo corrections and simplification of the writing, and are evident from the track changes, we do not document them below.